# Profiling of Epigenetic Features in Clinical Samples Reveals Novel Widespread Changes in Cancer

**DOI:** 10.3390/cancers11050723

**Published:** 2019-05-24

**Authors:** Roberta Noberini, Camilla Restellini, Evelyn Oliva Savoia, Francesco Raimondi, Lavinia Ghiani, Maria Giovanna Jodice, Giovanni Bertalot, Giuseppina Bonizzi, Maria Capra, Fausto Antonio Maffini, Marta Tagliabue, Mohssen Ansarin, Michela Lupia, Marco Giordano, Daniela Osti, Giuliana Pelicci, Susanna Chiocca, Tiziana Bonaldi

**Affiliations:** 1Department of Experimental Oncology, IEO, European Institute of Oncology IRCCS, 20139 Milan, Italy; camilla.restellini@ieo.it (C.R.); evelynoliva.savoia@ieo.it (E.O.S.); francesco.raimondi@ieo.it (F.R.); lavinia.ghiani@ieo.it (L.G.); daniela.osti@ieo.it (D.O.); Giuliana.pelicci@ieo.it (G.P.); susanna.chiocca@ieo.it (S.C.); 2Novel Diagnostics Program, IEO, European Institute of Oncology IRCCS, 20141 Milan, Italy; giovanna.jodice@ieo.it (M.G.J.); giovanni.bertalot@ieo.it (G.Be.); 3Biobank for Translational Medicine Unit (B4MED), Department of Pathology and Laboratory Medicine, IEO, European Institute of Oncology IRCCS, 20141 Milan, Italy; giuseppina.bonizzi@ieo.it (G.Bo.); maria.capra@ieo.it (M.C.); 4Division of Pathology, IEO, European Institute of Oncology IRCCS, 20141 Milan, Italy; fausto.maffini@ieo.it; 5Division of Otolaryngology Head & Neck Surgery, IEO, European Institute of Oncology IRCCS, 20141 Milan, Italy; marta.tagliabue@ieo.it (M.T.); Mohssen.ansarin@ieo.it (M.A.); 6Unit of Gynecological Oncology Research, IEO, European Institute of Oncology IRCCS, 20141 Milan, Italy; michela.lupia@ieo.it (M.L.); Marco.giordano@ieo.it (M.G.)

**Keywords:** epigenetics, histone post-translational modifications, mass spectrometry, proteomics, cancer, cell cycle, cell culture

## Abstract

Aberrations in histone post-translational modifications (PTMs), as well as in the histone modifying enzymes (HMEs) that catalyze their deposition and removal, have been reported in many tumors and many epigenetic inhibitors are currently under investigation for cancer treatment. Therefore, profiling epigenetic features in cancer could have important implications for the discovery of both biomarkers for patient stratification and novel epigenetic targets. In this study, we employed mass spectrometry-based approaches to comprehensively profile histone H3 PTMs in a panel of normal and tumoral tissues for different cancer types, identifying various changes, some of which appear to be a consequence of the increased proliferation rate of tumors, while others are cell-cycle independent. Histone PTM changes found in tumors partially correlate with alterations of the gene expression profiles of HMEs obtained from publicly available data and are generally lost in culture conditions. Through this analysis, we identified tumor- and subtype-specific histone PTM changes, but also widespread changes in the levels of histone H3 K9me3 and K14ac marks. In particular, H3K14ac showed a cell-cycle independent decrease in all the seven tumor/tumor subtype models tested and could represent a novel epigenetic hallmark of cancer.

## 1. Introduction

Traditionally, cancer has been considered as the result of the accumulation of genetic defects, such as mutations and copy number changes. However, striking evidence has now shown how epigenetic changes also contribute to cancer progression [1]. Epigenetic features include histone post-translational modifications (PTMs), DNA methylation, and nucleosome positioning, which all concur to the regulation of gene expression, ultimately determining the fate of the different cell types of an organism starting from the same DNA sequence. The disruption of epigenetic mechanisms can lead to altered gene expression and cellular transformation, which play a crucial role during the initiation and progression of cancer. 

Histone modifications represent a vast catalogue of combinatorial events that occur on the tail of histone proteins, the protein component of chromatin. The list of histone PTMs is ever expanding and includes well characterized modifications (e.g., acetylation and methylation [2]) as well as novel modifications (e.g., serotonylation [3]). The type, location, and combination of histone PTMs contribute to the regulation of the underlying DNA by affecting chromatin compaction and accessibility, and by generating binding sites for the recruitment of effector proteins that mediate downstream processes [1,2]. Specific modified residues mark genomic regulatory regions and directly influence gene expression. For instance, active promoters are enriched in histone H3 (H3) K4me3, inactive promoter regions are enriched in H3K9me3 or H3K27me3, and active enhancers are marked by H3K27ac (Hawkins et al., 2011; Hon et al., 2009; Mills, 2010), just to name a few of the most characterized histone PTMs. 

Histone modification levels are regulated by the expression and activity of histone modifying enzymes (HMEs), which depose and remove specific modifications. Aberrations in the expression and localization of HMEs, as well as mutations in their sequences, have been linked with different types of cancer [4]. Such aberrations often result in the disruption of normal histone PTMs patterns and can impact on gene expression, leading to the inappropriate activation of oncogenic pathways or inactivation of tumor suppressors.

Besides epigenetic changes occurring in specific tumors, epigenetic features have been recognized as general hallmarks of cancer, including genome-wide hypomethylation, site-specific hypermethylation of CpG island promoters [5,6], and histone PTM changes. In 2005, Fraga and colleagues employed immunodetection, high-performance capillary electrophoresis, and top-down mass spectrometry (MS) to analyze histone H4 PTMs in a large panel of tumoral and non-tumoral cell lines, as well as in two human primary tumors and their corresponding normal counterpart, finding a reduction of monoacetylated K16 and trimethylated K20 in cancerous samples. These losses appeared to occur early during the tumorigenic process and to become more accentuated in advanced stages, as the authors demonstrated by studying a mouse model of skin cancer progression [7]. Loss of these two histone marks was observed particularly in DNA repetitive regions, which are typically undermethylated in cancer [7]. By demonstrating that alterations in histone PTMs can be associated with cancer, this study represented an important proof of concept for the use of MS-based approaches to study the epigenetic profile of cancer cells. Following studies involving the profiling of histone H3 and H4 modification in cell lines confirmed the overall decreasing trend of H4K16ac and H4K20me3 and showed other potential cancer-related epigenetic changes, such as an increase of H3K27me2/me3 [8,9]. A few other histone PTM changes associated with specific tumors have been reported [10,11], but no other studies addressing, in a comprehensive manner, histone modification changes as general hallmarks of cancer, or as hallmarks of specific tumors, have been performed. In addition, most of the studies performed so far involved testing only cell lines, which are known to undergo a dramatic epigenetic rewiring compared with their tumor of origin [12,13]. As a consequence, results obtained in cultured models should be taken with caution.

In this study, we took advantage of a battery of MS-based approaches, recently developed in our laboratory, for the analysis of histone PTMs from different types of primary samples, including formalin-fixed paraffin-embedded (FFPE) tissues, fresh-frozen, and optimal cutting temperature (OCT) frozen tissues [13,14,15], to comprehensively profile histone H3 lysine methylations and acetylations in different human cancer and normal tissues, identifying a decrease of H3K14 acetylation as a common change in tumoral tissues, as well as tumor- and subtype-specific epigenetic changes. 

## 2. Results and Discussion

### 2.1. MS-Based Profiling of Normal and Tumor Tissues

In recent years, mass spectrometry has emerged as the most useful tool to dissect histone PTMs in a comprehensive, unbiased, and quantitative manner. With the goal of identifying histone PTM marks that could represent general features of cancer, but also tumor- and subtype- specific changes occurring in tumor cells, we employed MS-based approaches to profile histone H3 lysine acetylations and methylations in tumoral and normal tissues for different cancer types. In particular, we analyzed breast cancer, ovarian cancer, and head and neck cancer samples, and compared them with their normal counterpart. For breast cancer, we considered two subtypes, luminal A-like and triple negative, that show very different features and can be considered as two different cancer types (Appendix A). Indeed, luminal A-like tumors show expression profiles similar to those of luminal mammary epithelial cells, express hormone receptors, have a low proliferation rate, and typically have a good prognosis. On the contrary, triple negative breast cancers, which usually show a basal-like phenotype and express genes associated with normal breast myoepithelial cells, do not express any molecular markers, have a higher proliferation rate, and are associated with a poor prognosis [16]. Luminal A-like and triple negative tumors were compared with the normal tissue surrounding the tumors (matched samples were used, when available) (Appendix A). Ovarian cancer specimens belonging to the high-grade serous histotype, which is the most frequent and aggressive form of the disease, were compared with non-neoplastic samples from patients undergoing adnexectomy for non-ovarian gynecological pathologies (Appendix A), while head and neck oropharyngeal cancers were compared with normal tonsils of patients undergoing surgery for other head and neck related pathologies (Appendix A).

Patient sample sources were FFPE, fresh-frozen or OCT frozen, depending on specimen availability (Appendix A), and ad hoc isolation protocols [13,14,15] were used for each type of starting material. Following extraction and enrichment, histones were processed through an in-gel digestion, which is ideal to dissect histone H3 modifications [17]. Following digestion into peptides, the samples were analyzed by liquid-chromatography (LC)-MS, using an established platform that allows quantifying up to 30 histone H3 PTMs from frozen tissues and 24 from FFPE tissues, where few modifications cannot be reliably quantified due to storage artifacts [15]. The quantification of differentially modified peptides was achieved by using a super-SILAC (Stable Isotope Labeling with Amino acids in Cell culture) spike-in standard, which consists in heavy labeled histones that are mixed with the histone preparations from clinical samples prior to the digestion and are subjected to the same processing steps, thus reducing experimental variability and increasing quantitation accuracy [15]. 

We found a number of changes in the comparison of different tumors and their corresponding normal tissues (Figure 1A,B). A few of them were specific to one tumor type, such as the decrease in the mono-acetylated form of the histone H3 18–26 peptide in luminal A-like tumors, or the decrease of H3K27me1 in ovarian cancer. The majority was instead common to two or more tumor types. Most of the changes observed in luminal A-like samples were also present in triple negative breast cancer samples, where additional changes could be observed (these included an increase of H3K9me3, H3K36me1, and H3K79ac, and a decrease of H3K27me3-containing peptides and H3K79me1/me2). This result is in agreement with what we have recently reported for histone H4 PTMs, where triple negative samples showed many more changes than luminal A-like samples, compared with normal tissues [18]. This finding is also consistent with the notion that luminal A-like samples have a lower proliferation rate, are less aggressive and typically more differentiated [19], and, as such, they may be more similar to normal tissues, compared with triple negative samples. Some changes were observed in all tumor types, except for luminal A-like samples, such as an increase of the methylated forms of H3K36, which was often paralleled by a decrease of H3K27me3. Typically, changes found in more than one tumor type were in the same direction, suggesting possible common underlying epigenetic mechanisms. The only exception was represented by the H3K27me2 mark, which was decreased in breast cancer subtypes and increased in ovarian cancer.

The most widespread changes occurred on the histone H3 9–17 peptide. We found a decrease of at least one peptide containing H3K14ac in all the tumor types tested (highlighted in blue in Figure 1A), which was paralleled by an increase of the non-acetylated forms of the 9–17 peptide. We therefore expanded the analysis of modifications on the 9–17 peptide to other panels of normal-tumor samples, which included luminal B-like breast cancers (Appendix A), an additional group of ovarian cancers (Appendix A), prostate cancers (Appendix A), and mouse glioblastoma samples (Figure 1C,D). The decrease of H3K14ac-containing peptides was confirmed in all these tumor samples (Figure 1C,D), as well as in a small panel of breast cancer patient samples belonging to the luminal A-like and triple negative subtypes, where homogeneous tumor populations were laser-microdissected from FFPE sections (from [14]) (Figure 1E). When comparing the average total levels of H3K14ac (given by the sum of H3K14ac, H3K9me1/K14ac, H3K9me2/K14ac, H3K9me3/K14ac, and H3K9ac/K14ac) in the normal-tumor pairs for all the tumor types tested, we found a highly significant decrease in tumor samples, with a paired *t*-test *p*-value of 0.00013 (Figure 1E). Of note, we did not observe a decrease of total H3K14ac in mouse glioblastoma, but only of H3K14ac in combination with H3K9ac. In three out of the seven tumor types tested, we also observed an increase of total H3K9me3 (given by the sum of H3K9me3 and H3K9me3K14ac, Figure 1E). Taken together, these results suggest that H3K14ac may represent a novel general hallmark of cancer, while other histone PTM changes may represent more specific epigenetic aberrations found in distinct tumor types. 

### 2.2. MS-Based Profiling of Cycling Cells

Since histone PTM abundances may change during the cell cycle [20,21] and tumor cells are typically more proliferative than normal cells, one possibility is that differing proportions of cycling cells in tumoral and normal samples could account for the observed histone PTM changes in tumors. Studies investigating histone PTMs levels during the cell cycle using MS-based approaches [20] or high-throughput imaging [21] reported discordant results, showing either a decrease or a slight increase of global acetylation and H3K27/K36 methylation, respectively, during the G2/M phase. Previously, we found a significant/close to significant correlation between the proliferation index Ki67 and H3K36me1/me2 and H3K27me3 in luminal breast cancer and glioblastoma tissues [13], suggesting that these marks may be influenced by the cell cycle. Here, we used the same strategy and correlated the L/H (light/heavy) ratios obtained for peptides that appear to change in the tumor, compared with normal tissues with the Ki-67 index of breast cancer samples from two different datasets, as follows: (1) the samples analyzed in this study (Figure 2A, left panels) and (2) the FFPE dataset containing breast cancer samples belonging to the four main subtypes from our previous publication ([15], Figure 2A, right panels). The analysis of the two datasets gave overall similar results. K14ac did not show any correlation, while K9me3 showed a positive correlation with the Ki-67 index. In addition, consistent with our previous observation, H3K27me3- and H3K36me1/me2-containing peptides correlated positively and negatively, respectively, with the proliferation index. These results would suggest that the differences observed in tumors compared with normal tissues for K27, K36, and K9 methylation, but not K14 acetylation, may be due to the increased tumor proliferation rates. However, this type of analysis does not allow for discriminating the effect of proliferation from other factors influencing histone PTM patters, including specific histone PTM differences existing among breast cancer subtypes [15].

To more specifically address the issue of histone PTM changes during the cell cycle, we profiled histone H3 PTMs in G2-M synchronized cells versus G1-S-synchronized cells (Appendix A), in MCF7 and MDA-MB-231 cells (luminal A and triple negative breast cancer cell lines, respectively) and the normal breast MCF10A cell line (Figure 2B,C and Appendix A). The H3K14ac mark decreased in the G2-M phase in MDA-MB-231 cells, but remained constant in MCF7 and even increased in MCF10A cells, suggesting that the decrease that we have observed in tumors is not a mere consequence of different proliferation rates (Figure 2B–D). Differently from the correlation analysis, H3K9me3 markedly decreased in all the cell lines analyzed in the G2-M phases, suggesting that the increase of this mark in tumors is also not cell-cycle dependent. Conversely, both from the correlation and the cell cycle analyses, the decrease of the H3K27me3 mark appeared to be linked with the cell cycle, in agreement with a previous study showing that H3K27me3 correlates with the terminal differentiation of proliferating myoblasts and is specific to permanently arrested cells [22]. Instead, despite correlating with the Ki-67 index, H3K36 methylation did not show any marked changes during the cell cycle. Because it is known that that H3K27 methylation antagonizes H3K36 methylation, and vice-versa [23,24,25], it is possible that the increase of K36 methylation may be a secondary long-term effect of the decrease of K27me3. This hypothesis would be consistent with the fact that these changes were not observed in luminal A-like breast cancer tumors, which have lower proliferation rates compared with other tumors. 

These results also have implications for our previously reported analysis of histone PTMs in breast cancer subtypes, where we found luminal A-like tumors to have lower levels of H3K27me3 compared to the other subtypes, while triple negative and HER2 (Human epidermal growth factor receptor 2) positive samples showed higher K9me3 levels [15]. Based on our novel findings, it appears like the different levels of the K27me3 mark in different breast cancer subtypes could be at least partially due to different proliferation rates, while H3K9me3 is confirmed as an interesting marker for breast cancer patient stratification, as it does not merely depend on proliferation. 

### 2.3. Expression of Histone Modifying Enzymes in Normal and Tumor Tissues

Alterations in histone PTM levels are often a consequence of the anomalous expression, mislocalization, or mutation of the HMEs responsible for their deposition and removal [26,27,28]. The most studied HMEs can be divided into four main classes, as follows: methyltransferases and demethylases, and acetyltransferases and deacetylases, which transfer/remove methylations or acetylations, respectively. Typically, methyltransferases and demethylases specifically act on one or few histone residues [27,29], while deacetylases and acetyltransferases are capable of acetylating and deacetylating multiple histone sites, showing lower substrate specificity. Nevertheless, some deacetylases and acetyltransferases also show a preference for a subset of lysine substrates [27,29]. 

We analyzed RNASeq data from the The Cancer Genome Atlas (TCGA) to extrapolate the differential expression levels of 88 known or putative HMEs, of which 17 acetyltransferases, 18 deacetylases, 32 methyltransferases, and 21 demethylases, in 16 tumor tissues, and compared them with the corresponding normal tissues. As expected, we found a large number of HMEs whose levels are altered in tumors, with 67 proteins showing a significant enrichment/depletion compared with normal tissues in at least one tumor type (Figure 3, Appendix A). Overexpression of histone deacetylases has been reported in many cancer patients [30]. Mutations, including deletions, amplifications, point mutations, and translocations of histone acetyltransferases have been described and, in some instances, altered expression levels have been observed in the absence of DNA alterations [31]. In our analysis, acetyltransferases and deacetylases showed complex patterns, with some of the enzymes displaying strong overexpression in some tumor types and downregulation in others, in accordance with numerous studies implicating histone acetyltransferases, both as oncogenes and tumor suppressors [32].

The following three were the most common changes observed in tumors at the level of acetyltransferases: KAT2B showed significantly lower levels in 14 out of 16 tumor types; NCOA1/SRC1/KAT13A was downregulated in 9 tumors, and KAT2A was upregulated in 11 tumors. KAT2A (also known as GCN5) and KAT2B (also known as PCAF) belong to the GNAT (Gcn5-related N-acetyltransferase) family of acetyltransferases and are transcription co-activators that work as part of large multi-subunit complexes, where they are present in a mutually exclusive manner [33]. KAT2A and KAT2B carry out many redundant, but also some specific, functions. In cancer, GCN5 acts in a oncogenic fashion, which is usually linked to its function as a coactivator of the c-MYC oncogene [34] and was previously found to be upregulated in urothelial cancer cell lines [34] and colon cancer human tissues [35]. Instead, KAT2B mostly behaves as a tumor suppressor. Indeed, it induces cell apoptosis in hepatocellular carcinoma cell lines [36,37], inhibits gastric cancer growth [38], suppresses lung adenocarcinoma progression [39], and is downregulated in various tumor types, including esophageal, breast, ovarian, colorectal, pancreatic cancers, and stomach cancer [40]. Since histone H3 K14 and K9 have been reported as the preferred sites for acetylation by KAT2A and KAT2B [33], it is possible that an altered equilibrium in the expression of these enzymes is responsible for the general decrease of H3K14 acetylation observed in tumors (Figure 1). NCOA1 may also play a role in this process, as it is known to interact with KAT2B and other acetyltransferases, influencing their activity [41]. Other acetyltransferases that act on H3K14 (ELP3, GTF3C4, GOT2, and KAT5) showed altered levels only in a few tumor types and may contribute to altered H3K14ac levels in specific contexts.

Concerning methylation, in our analysis histone demethylases were mostly upregulated in tumors, while methyltransferases were partly down- and partly up-regulated. Interestingly, many methyltransferases acting on the histone H3K9 residue were upregulated (SUV39H1, SUV39H2, EHMT2, EHMT1, and SETDB1), in agreement with the increase in H3K9me3 observed in several of the tumor types that we tested (Figure 1). As previously reported [42], we found the EZH2 to be overexpressed in many different types of cancer. Conversely, three of the methyltransferases acting on K36 were generally downregulated (SETD2, SETD3, and SETMAR), while H3K36 demethylases (KDM2A, KDM2B, KDM4A, and KDM4B) were upregulated in various tumors. These results would be consistent with an increase of H3K27 methylation and a decrease of H3K36 methylation. However, in our PTM analysis, we found the opposite result, namely a general decrease of H3K27me3 and an increase of H3K36me1/me2 (Figure 1). This indicates that other factors play a more predominant role in determining the levels of these two modifications. While it has been reported that HME transcript levels are generally predictive of histone PTM abundances, it has been also shown that several PTMs are regulated independently of the levels of their modifying enzyme expression [9]. 

Several other mechanisms can contribute to histone PTM changes. Based on our results, the increased proliferation of tumor cells could be one of them and, in particular, could contribute to the changes found in H3K27me3 levels. In addition, the activity of HMEs could be affected regardless of changes in their expression levels. HMEs are known to be part of multi-subunit complexes, where more than one enzyme is often present [43], influencing their activities. The network of interactions of HME shown in Figure 4A (where enzymes belonging to different clusters based on network modularity optimization [44] are marked by different colors) clearly depicts the complexity of the interactions of these enzymes. Interestingly, when taking into account the enzyme differential expression in tumor compared with normal tissues (Figure 4B and Appendix A show the interaction networks for three of the tumor types where we profiled histone PTMs), it appears that like members of the same clusters behave similarly in terms of differential expression. For instance, the majority of the downregulated enzymes belong to cluster 2 and the upregulated enzymes to clusters 1, 3, and 6, suggesting possible aberrations at the levels of complex interactions, rather than single enzymes.

In addition, mutations can affect, positively or negatively, the activity of HMEs. Chromatin modifiers have been reported as one of the most heavily mutated class of proteins in cancer [46]. Accordingly, the mutational analysis of the patient samples present in the TCGA pan-cancer Atlas study [47] shows that 20%–90% of each tumor type presents at least one mutation in one of the HMEs, catalyzing the deposition and removal of methylations or acetylations (Figure 5A). The tumor types containing the highest number of mutations in HMEs are bladder cancer (BLCA), esophageal cancer (ESCA), and melanoma (SKCM), where at least 80% of the patients contain at least one mutation (which include missense mutations, deep deletions, and amplifications) and 35%–55% contain multiple mutations. When looking at individual enzymes (Appendix A), we found variable frequencies of alterations, as well as various proportions of different types of mutations. While in most of the cases, both amplifications and deletions of the same enzymes can occur in different tumors, or even in the same tumor type, some enzymes show a clear predominance of amplifications (e.g., SIRT2, SETDB1, KDM2A; KDM5A; KAT7) or deletions (e.g., ELP3; HDAC2; HDAC4; HDAC10, SIRT3; SETDB2) in all/most of the tumor types analyzed, suggesting a possible general oncogenic or tumor suppressor role, respectively. Among them, NCOA1, KAT7, and SETDB1 have indeed been reported as oncogenes, according to the Cancer Census Genes [48]. Interestingly, in some instances, enzymes with similar functions, such as SETDB1 and SETDB2 (methyltransferases acting on H3K9), showed diverging mutation types (amplifications for SETDB1 and deletions for SETDB2) (Figure 5B). These results are in agreement with the changes in expression levels of these two enzymes, where we observed an increase of SETDB1 in 13 out of 16 tumor types analyzed and a decrease of SETDB2 in 8 tumor types (Figure 3). These results suggest that a gain of function for SETDB1 and a loss of function for SETDB2 may be advantageous for tumors. In line with this hypothesis, SETDB1 promotes the growth of colorectal cancer [49] and hepatocellular carcinoma cells [50] and is an unfavorable prognostic marker in renal and liver cancer. On the contrary, SETDB2 is a favorable prognostic marker in renal cancer ([51] www.proteinatlas.org). However, in stomach cancer, one of the few tumor types where SETDB2 is amplified and not deleted, this enzyme has been shown to have a tumor promoting role [52]. These examples show how a mutational analysis can provide clues on the role of HMEs in cancer. Various deacetylases and few acetyltransferases also show predominant deletions or amplifications. Among them, one acetyltransferase specific for H3K14–ELP3 contains almost exclusively deletions. While this enzyme does not show altered levels of expression (Figure 3), its reduced activity may contribute to the reduced levels of H3K14ac in a portion of the tumors.

When testing the significance of mutation enrichment in each cancer type, we found a total of 19 unique and significantly mutated HMEs (MutSig2CV adjusted *p*-value < 0.01) in 18 TCGA cancer types (Figure 5C). EP300, SETD2, and KDM6A were found significantly mutated in multiple tumor types, while mutations in other enzymes were tumor-specific (Figure 5C). Among them was the above-mentioned SETDB1, which was associated with stomach cancer. The single most recurrent mutated enzyme is NSD1 in head and neck cancer, where loss-of-function mutations in a subset of patients have been reported to promote a favorable response to chemotherapy [53].

Finally, we investigated the relationships between mutation status and differential expression of HMEs, finding six unique HMEs (EHMT2, EZH2, HDAC4, KDM5C, KDM6A, and SETD2) both significantly mutated and differentially regulated in 5 cancer types (Colon adenocarcinoma (COAD), kidney renal clear cell carcinoma (KIRC), lung adenocarcinoma (LUAD), prostate adenocarcinoma (PRAD), and stomach adenocarcinoma (STAD)), although we found no statistical significance of co-occurrence between the two events (Dataset S3).

### 2.4. Comparison of Histone PTMs in Normal and Tumor Cell Lines

In a recent study, we investigated the effects of the culture conditions on histone PTM patterns, revealing an extensive and systematic rewiring of histone H3 marks in several normal and tumoral cell culture models (both 2D and 3D), which included breast and ovarian cancer, glioblastoma, and normal breast and brain. Such rearrangement causes the disappearance/reduction of histone PTM differences among cancer types and is relevant to the point that differences between tissues and cell lines are more marked than those among cancer types [13]. Here, we show that the rewiring of histone H3 PTMs in culture is also generally predominant, with respect to changes between normal and tumor samples, as shown by the principal component analysis (PCA) of normal and tumor tissues and cell lines for luminal A-like and triple negative breast cancer and prostate cancer (Figure 6A).

We also tested whether individual histone PTM changes found in tumors relative to normal tissues were retained in cell lines by comparing luminal A and triple negative breast cancer cell lines with normal breast cell lines. With regards to the two published hallmarks of cancer [7], we found a decrease of H4K20me3 in both subtypes, but no change for H4K16ac (Figure 6B). This result is somewhat surprising, given that H4K20me3 and H4K16ac were initially identified by profiling a large set of cancer and normal cell lines. However, cell lines belonging to different tumor/tissue types were pooled, so it is possible that not all the tumors/tumor subtypes show the same change and that the decrease of H4K16 is not as widespread in tumors as previously thought.

No change was detected at the levels of H3K27me3 or H3K36me1/me2, possibly because the cancer cell lines do not have a proliferation rate higher than the normal cells (the doubling times for MCF7, MDA-MB-231, and MCF10A are 41, 43, and 24 hours, respectively). Additionally, in luminal A cells, there was a decrease of the tetra-acetylated form of the histone H4 peptide 4–17, contrary to the increase that we have previously observed in luminal A-like tumor tissue [18]. The decrease of H3K14ac-containing peptides was lost in both breast cancer subtypes, while the increase of H3K9me3 was somewhat maintained (Figure 6B–D). We also profiled the differentially modified forms of the H3 9–17 peptide in prostate cancer cell lines and mouse glioblastoma long-term primary neurosphere cultures, compared with non-tumoral cells (Figure 6C,D). No change could be detected in tumor prostate cancer cell lines, while many differences were present in glioblastoma, compared with brain cultures. It must be noted that the biological replicates for the neurosphere cultures were cells originating from the same mouse and collected at different passages, which likely explains the lower variability of the measurements and the higher significance of the differences observed. Compared with the differences found in mouse glioblastoma and normal tissues, the increase in H3K9me3/K14ac was maintained in culture (but the total K9me3 decreased in tumor cell lines), while the marked decrease of H3K9ac/K14ac was lost, possibly because acetylations on the 9–17 histone H3 peptide are highly affected by culture conditions [13].

Taken together, these results highlight once more how cell lines retain some, but not all, of the features of the tissue samples they derive from and suggest that some of the cancer-induced epigenetic changes are lost/reduced in culture conditions, although an analysis comparing matched tissues and derived cell lines would be required for a definitive conclusion. This issue must be taken into account when looking for epigenetic cancer biomarkers or investigating epigenetic mechanisms in cell lines or other long-term primary cultures.

## 3. Materials and Methods 

### 3.1. Tissue Specimens 

Human cancer and normal samples were obtained from patients undergoing surgery at the European Institute of Oncology (Milan). Breast cancer specimens or normal breast tissues surrounding the tumors were obtained from patients undergoing surgery for the removal of clinically confirmed neoplasia. Matched normal-tumor samples were collected, when possible, and are indicated by the same sample number. For breast cancer samples, the levels of hormone receptors, Her-2 and Ki-67 were ascertained by immunohistochemistry [9]. Breast cancer subtypes were defined as follows: Luminal A-like: ER (estrogen receptor) and/or PgR(progesterone receptor)(+), HER2 (Human epidermal growth factor receptor 2)(−), Ki67 < 20%; luminal B-like: ER and/or PgR(+), HER2(−), Ki67 ≥ 20; triple negative: ER, PgR, and HER2(−), irrespective of Ki67 score; HER2 positive: HER2(+), irrespective of ER, PgR, or Ki67. ER/PgR positivity was defined as ≥1% of immunoreactive neoplastic cells and HER2 positivity was defined as >10% of neoplastic cells with strong and continuous staining of the cell membrane (3+ by immunohistochemistry) and/or amplified by in situ hybridization techniques, in accordance to the ASCO (American Society of Clinical Oncology)/CAP (College of American Pathologists) guidelines. Normal and tumor samples were selected and evaluated by a trained pathologist. Samples were with infiltrating carcinoma were selected to have a tumor cellularity of at least 50%, as assessed by hematoxylin and eosin (H&E) staining. Specimens with in situ carcinoma areas, large necrosis areas, and massive flogistic infiltration were discarded. Healthy ovarian tissues were obtained from patients undergoing adnexectomy for non-ovarian gynecological pathologies. Prostate cancer and normal prostate tissues adjacent the tumors were obtained from patients undergoing prostatectomy for the treatment of prostate adenocarcinomas. Normal head and neck tissues were obtained from tonsils of patients undergoing surgery for other head and neck related pathologies. Cancer and normal samples were obtained from the Biobank for Translational Medicine Unit (B4MED) of the European Institute of Oncology. Sample collection by the Biobank, in the presence of patient consent, was approved by the Ethical Committee of the European Institute of Oncology on 6 June 2011 and the samples can be used for retrospective studies without any further approval by the Ethical Committee [54]. For storage, samples were collected and snap frozen in liquid nitrogen, frozen in optimal cutting temperature compound (OCT), or fixed overnight in 4% formalin and embedded in paraffin. The list of patient samples analyzed in this study and the storage methods used are summarized in Appendix A.

### 3.2. Cell Lines and Primary Cultures 

Cell lines were grown in the media indicated in Appendix A. All growth media were supplemented with antibiotics (100 U/mL penicillin and 100 μg/mL streptomycin) (Thermo Fisher Scientific, Waltham, MA, USA) and 2 mM l-glutamine (Thermo Fisher Scientific, Waltham, MA, USA). Normal murine neural stem/progenitor cells were isolated from the sub-ventricular zone of adult C57BL/6J mice. Tumoral murine GL261 stem/progenitor cells were derived from C57BL/6J mice harboring GL261 glioma. Both murine normal and tumoral stem/progenitor cells were expanded as floating neurospheres and maintained in DMEM (Dulbecco’s Modified Eagle Medium)/ Ham’s F-12 medium (1:1) supplemented with B-27 (Thermo Fisher Scientific, Waltham, MA, USA), 20 ng/mL human epidermal growth factor (EGF), 10 ng/mL human basic fibroblast growth factor (bFGF) (PeproTech, London, UK), and 0.0002% heparin (Merck KGaA, Darmstadt, Germany). The analysis of normal and tumor mouse neurospheres was performed on cells derived from the same mouse, which were collected at different passages. Cell lines and primary cultures were grown in a humidified 37 °C incubator with 5% CO2.

### 3.3. Mouse Tissues

An immunocompetent model of murine glioblastoma (GBM) was established through intracranial implantation of murine GL261 neurospheres into C57BL/6J mice. Mice harboring GL261 brain tumors were maintained until the development of neurological signs. Then, the brain was removed, fixed overnight in 4% formalin, and embedded in paraffin. Experimental procedures involving animals complied with the Guidelines of the Italian National Institute of Health and were approved by the Institutional Ethical Committee.

### 3.4. Cell Synchronization

MCF7 and MCF10A cells were synchronized in the G1/S phase by incubating subconfluent cells in growing media for 38 and 24 hours, respectively, supplemented with 5 mM thymidine. MDA-MB-231 was synchronized in the G1/S phase by performing a double thymidine block consisting of 18 h of incubation in growing medium supplemented with 3.5 mM thymidine, 9 hours of release, and 24 hours of re-incubation with 3.5 mM thymidine. To arrest the cells in the G2/M phase, subconfluent MCF7, MCF10A, and MDA-MB-231 were incubated for 24 hours in growing media supplemented with 5 mM thymidine. Cells were then released for three hours and incubated for 20 hours (MDA-MB-231 and MCF7 cells) or 15 hours (MCF10A) in a medium containing 100 ng/mL nocodazole. Cell-cycle profiles of synchronized cell lines were verified by flow cytometry, according to propidium iodide content. The percentage of cells in the different phases of the cell cycle was calculated with the FlowJo software (Appendix A). The percentage of cells treated with nocodazole in the G2/M phase ranged from 49% (for MCF7) to 90% (for MDA-MB-231), but was always much higher than the percentage of G2/M cells after thymidine treatment, which ranged between 5.0% and 5.8%. The difference in the percentages of cells could affect the amplitude of histone PTM changes, but not the direction of changes between G2/M and G1/S phases and, as such, should not alter the overall result of the analysis. 

### 3.5. Histone Enrichment

Histones were purified from breast cancer cell lines through nuclei isolation on a sucrose cushion, followed by acidic extraction, as previously described [15], while they were enriched from the other cell lines and primary cells by resuspending 0.5–2 × 10^6^ cells in 1 ml of Phosphate-buffered saline (PBS) containing 0.1% Triton X-100 (Merck KGaA, Darmstadt, Germany) and protease inhibitors. Nuclei were isolated through a 10 min centrifugation at 2300× *g*, resuspended in the same buffer containing 0.1% sodium dodecyl sulfate (SDS), and incubated for few minutes at 37 °C in the presence of 250 U of benzonase (Merck KGaA, Darmstadt, Germany) to digest nucleic acids. To enrich histones from frozen tissues, at least 20 mg of tissue were thawed on ice, cut with scissors, and homogenized in 1 ml of PBS+Triton using a Dounce homogenizer. The homogenate was filtered through a 100 μm cell strainer and nuclei were isolated and lysed as described for primary cells. For optimal cutting temperature (OCT) frozen samples, the same procedure was used, after removing OCT from eight 10 μm-thick tissue sections with three washes in 70% ethanol, one in water, and two in PBS. Histones were isolated from formalin-fixed paraffin-embedded (FFPE) tissues using the PAT-H-MS (pathology tissue analysis of histones by mass spectrometry) protocol, as recently described [28]. Briefly, four 10 μm-thick tissue sections were de-paraffinized and rehydrated using standard procedures. Tissue samples were homogenized by sonication, in 200 μL of 20 mM Tris pH 7.4 containing 2% SDS, and proteins were extracted and de-crosslinked at 95 °C for 45 min and 65 °C for 4 h. Histones were obtained from paraffin-embedded whole mouse brains, harboring an orthotopic model for murine glioblastoma (GBM), from 7–10 μm-thick H&E stained sections that were subjected to manual macrodissection to isolate either the tumor or the normal brain tissue prior to PAT-H-MS [14]. Of note, few modified histone peptides, including H3K18me1 and methylations on H3K79, cannot be profiled in FFPE tissues, due to the insurgence of artifacts likely caused by formalin fixation [15]. The yield of histones deriving from the different purification protocols was estimated by SDS-PAGE (polyacrylamide gel electrophoresis) gel by comparison with known amounts of recombinant histone H3.1 (New England Biolabs, Ipswich, MA, USA), following protein detection with colloidal Coomassie staining (Expedeon, San Diego, CA, USA). 

### 3.6. Super-SILAC 

MDA-MB-231, MDA-MB-468, MDA-MB-453, and MDA-MB-361 breast cancer cells lines were grown in SILAC-DMEM (Thermo Fisher Scientific, Waltham, MA, USA), supplemented with 2 mM L-glutamine, 146 mg/L of lysine (Merck KGaA, Darmstadt, Germany), 84 mg/L L-13C615N4-arginine (Arg-10, Merck KGaA, Darmstadt, Germany), 10% dialyzed serum (Thermo Fisher Scientific, Waltham, MA, USA) and penicillin/streptomycin for at least 8 doublings to obtain complete labeling with heavy-labeled amino acids. Histones were isolated as previously described [3], mixed in equal amounts, lyophilized, and stored at −80 °C until use.

### 3.7. Histone Digestion

About 5–10 μg of histones per run per sample were mixed with an approximately equal amount of super-SILAC mix. For in-gel digestions, bands corresponding to histone H3 were excised, chemically alkylated with D6-acetic anhydride (Merck KGaA, Darmstadt, Germany), and in-gel digested with trypsin (the combination of chemical alkylation and trypsin digestion generates an “Arg-C-like” digestion) [1]. For the analysis of histone H4 PTMs, shown in Figure 6, an Arg-C in-solution digestion was performed, according to the manufacturer’s protocol (Merck KGaA, Darmstadt, Germany), overnight at 37 °C. All samples were desalted on handmade StageTips, as previously described [1].

### 3.8. LC-MS/MS Analysis of Histone PTMs

Peptide mixtures were separated by reversed-phase chromatography on an in-house-made 25-cm column (inner diameter 75 μm, outer diameter 350 μm outer diameter, 1.9 μm ReproSil, Pur C18AQ medium (Thermo Fisher Scientific, Waltham, MA, USA)), using a ultra-nanoflow high-performance liquid chromatography (HPLC) system (EASY-nLC™ 1000, Thermo Fisher Scientific, Waltham, MA, USA) or EASY-Spray columns (Thermo Fisher Scientific, Waltham, MA, USA), 25/50 cm long (inner diameter 75 µm, PepMap^TM^ C18, 2 µm particles), which were connected online to a Q Exactive HF instrument (Thermo Fisher Scientific, Waltham, MA, USA) through a Nanospray Flex™ or an EASY-Spray™ Ion Sources (Thermo Fisher Scientific, Waltham, MA, USA), respectively. Solvent A was 0.1% formic acid (FA) in ddH_2_O and solvent B was 80% ACN (acetonitrile) plus 0.1% FA. Peptides were injected in an aqueous 1% Trifluoroacetic Acid (TFA) solution at a flow rate of 500 nL/min and were separated with a 50 to 100 min linear gradient of 0%–40% solvent B for in-gel digested samples and a 60 to 70 min linear gradient of 0%–40% solvent B for Arg-C digested samples, at a flow rate of 250 nL/min. The Q Exactive HF instrument was operated in the data-dependent acquisition (DDA) mode to automatically switch between full scan MS and MS/MS acquisition. Survey full scan MS spectra (m/z 300–1350) were analyzed in the Orbitrap detector with a resolution of 60,000 at m/z 200. The 10 most intense peptide ions, with charge states comprised between 2 and 4, were sequentially isolated to a target value for MS1 of 3 × 10^6^ and fragmented by HCD (Higher-energy C-trap dissociation) with a normalized collision energy setting of 28%. The maximum allowed ion accumulation times were 20 ms for full scans and 80 ms for MS/MS and the target value for MS/MS was set to 1 × 105. The dynamic exclusion time was set to 10 sec and the standard mass spectrometric conditions for all experiments were as follows: Spray voltage of 1.7–1.8 kV, no sheath, and auxiliary gas flow.

### 3.9. Histone PTM Data Analysis

Acquired raw data were analyzed using the integrated MaxQuant software v.1.5.2.8 (Max Planck Institute of Biochemistry; https://maxquant.org/), which performed peak list generation and protein identification using the Andromeda search engine. The Uniprot HUMAN_histones 1502 database was used for histone peptide identification. Enzyme specificity was set to Arg-C. The estimated false discovery rate (FDR) of all peptide identifications was set at a maximum of 1%. The mass tolerance was set to 6 ppm for precursor and fragment ions. One missed cleavage was allowed and the minimum peptide length was set to 6 amino acids. Variable modifications for in-gel digestions included lysine D6-acetylation (+45.0294 Da), lysine monomethylation (+59.0454, corresponding to the sum of D6-acetylation (+45.0294) and monomethylation (+14.016 Da), dimethylation (+28.031 Da), trimethylation (+42.046 Da), and lysine acetylation (+42.010 Da). To reduce the search time and the rate of false positives, which increase with increasing the number of variable modifications included in the database search, the raw data were analyzed through multiple parallel search jobs, setting different combinations of variable modifications, as previously described [15]. Variable modifications for in-solution Arg-C digestions were lysine monomethylation (+14.016 Da), dimethylation (+28.031 Da), trimethylation (+42.046 Da), and acetylation (+42.010 Da). Peptides with Andromeda scores less than 60 and localization probability scores less than 0.75 were removed. Identifications, retention times, and elution patterns of isobaric peptides were used to guide the manual quantification of each modified peptide using QualBrowser version 2.0.7 (Thermo Fisher Scientific, Waltham, MA, USA). Site assignment was evaluated using QualBrowser (Thermo Fisher Scientific, Waltham, MA, USA) and MaxQuant Viewer (Max Planck Institute of Biochemistry https://maxquant.org/) from MS2 spectra. Extracted ion chromatograms (XICs) were constructed for each doubly/triply charged precursor, based on its m/z value, using a mass tolerance of 10 ppm and a mass precision up to four decimals. For each histone modified peptide, the % relative abundance (%RA) was estimated by dividing the area under the curve (AUC) of each modified peptide for the sum of the areas corresponding to all the observed forms of that peptide and multiplying by 100. For SILAC experiments, Arg10 was selected as a heavy label (multiplicity = 2) in MaxQuant. The heavy form of each modified peptide was quantified from its XIC and the relative abundance was quantified. Heavy peptides without a light counterpart were not considered for quantification. The AUC values for all the samples analyzed are reported in Datasets S1–S2. The mass spectrometry proteomics data have been deposited to the ProteomeXchange Consortium [8] via the PRoteomics IDEntifications (PRIDE) partner repository with the dataset identifiers PXD013288 and PXD013311.

### 3.10. Analysis of TCGA RNAseq Expression Data

We obtained raw read counts for each gene in all cancer types released from TCGA (The Cancer Genome Atlas) before 26 February 2016. To avoid unreliable results, we selected 16 cancer types with at least 10 pairs of tumor-normal tissue matches (BLCA: Bladder Urothelial Carcinoma; BRCA; Breast invasive carcinoma; COAD: Colon adenocarcinoma; ESCA: Esophageal carcinoma; HNSC: Head and Neck squamous cell carcinoma; KICH: Kidney Chromophobe; KIRC: Kidney renal clear cell carcinoma; KIRP: Kidney renal papillary cell carcinoma; LIHC: Liver hepatocellular carcinoma; LUAD: Lung adenocarcinoma; LUSC: Lung squamous cell carcinoma; PRAD: Prostate adenocarcinoma; READ: Rectum adenocarcinoma; STAD: Stomach adenocarcinoma; THCA: Thyroid carcinoma; UCEC: Uterine Corpus Endometrial Carcinoma). Of note, it was not possible to find matched normal samples for ovarian cancer and glioblastoma, which therefore could not be included in the analysis. We used Deseq2 [55] for differential gene expression analysis in each cancer type and adjusted *p*-values (padj) for multiple testing by the Benjamini and Hochberg method (https://www.jstor.org/stable/2346101). We considered only values with significance padj < 0.01. To check for relationships between mutation status and differential expression of HMEs, we derived the list of significantly mutated genes through the MutSig2CV approach [56], available from the Firebrowse server (http://firebrowse.org/) for each TCGA cohort. We then checked whether significantly regulated HMEs (DESEQ2 adjusted *p*-value < 0.01) were also significantly mutated (MutSig2CV adjusted *p*-value < 0.01) through a Fisher’s exact test. We generated stacked bar plots summarizing mutation type and frequency through the cBio portal (https://www.cbioportal.org/) [57], using the TCGA PanCan 2018 cohort [47].

### 3.11. Statistical Analysis

Statistical testing and data display for histone PTMs were carried out using the Perseus software [10]. Changes in single histone modifications among two groups were evaluated by Student’s T test, with a Student’s *t*-test *p*-value of 0.05 or 0.1, which were indicated by a darker and lighter color, respectively, in Figure 1A,D; Figure 2C; and Figure 5A,C. A *p*-value < 0.05 with only two sample measurements was also indicated by lighter colors. Given the exploratory nature of this study, no correction for multiple comparison testing was applied [58]. *P*-values for each comparison are reported in Datasets S1–S2. Normalized L/H ratios, defined as L/H ratios of relative abundances normalized over the average value across samples, were visualized and clustered with correlation distance and average linkage as parameters. Principal component analyses (PCA) were performed by removing samples with more than 10 missing values and peptides that had no valid values in one category, or more than 5 missing values. The remaining missing values were substituted with the average value across all the samples analyzed. A Benjamini–Hochberg FDR of 0.05 was used as a cutoff method.

## 4. Conclusions

In this study, we investigated epigenetic aberrations occurring in tumors, using the following three different strategies: (1) profiling histone H3 PTM changes in normal and tumor tissues by MS-based methods, (2) verifying whether these changes are due to increased proliferation of tumor cells and whether they are maintained in culture models, and (3) analyzing the differential gene expression levels and the mutations of HMEs using publicly available datasets. In tissues, we found that some changes occurring at the level of histone PTMs and HMEs were specific to a single or a few tumor types, but other changes were much more widespread, suggesting the existence of common epigenetic aberrant mechanisms occurring in cancer. Among them, we found a general decrease of the H3K14ac mark in various tumor types, a change that is cell-cycle independent and that could thus represent a potential novel epigenetic hallmark of cancer. It will be important in the future to validate this finding on larger cohorts of patients, and additional tumor types, in order to verify whether the decrease of H3K14 in tumors can be generalized to other cancer types, in addition to the ones tested in this study. Furthermore, another crucial aspect that will have to be addressed is tissue heterogeneity. By analyzing whole tissue sections with a cut-off tumor cellularity of 50%, we may have underestimated some of the differences found in tumor cells compared with normal cells, but we may also have neglected the contribution of non-tumoral cells present in the samples. By confirming the decrease of H3K14ac in tumor cell populations that were laser microdissected from luminal A-like and triple negative tumors, we confirmed that this change is likely specific to tumor cells. However, we did not investigate the contribution of different cell populations in the normal tissues (for instance we did not distinguish luminal and myoepithelial breast cells or take into account the presence of immune cells), the level of differentiation of the tumors, or differences present in different areas of the tumors. All these considerations also apply to the previously identified hallmarks of cancer (loss of H4K20me3 and H4K16ac) [7], which were discovered by profiling cell lines and validated on a very limited number of tumor types. 

The correlation of the histone PTM profiling and gene expression profiling data showed that few histone PTM changes correlated with changes in their corresponding HMEs, while others did not. This indicates that the aberrant histone PTM levels may be the result of multiple mechanisms, which include the aberrant levels of HMEs, as well as their mutation and aberrant turnover rates, the altered function of multi-subunit complexes of which HMEs are part, and the increased proliferation rate of tumor cells. 

Taken together, our results unraveled novel epigenetic features altered in one or more tumor types, which will be worth investigating further in the future as possible tumor epigenetic markers and as potential therapeutic targets. 

## Figures and Tables

**Figure 1 cancers-11-00723-f001:**
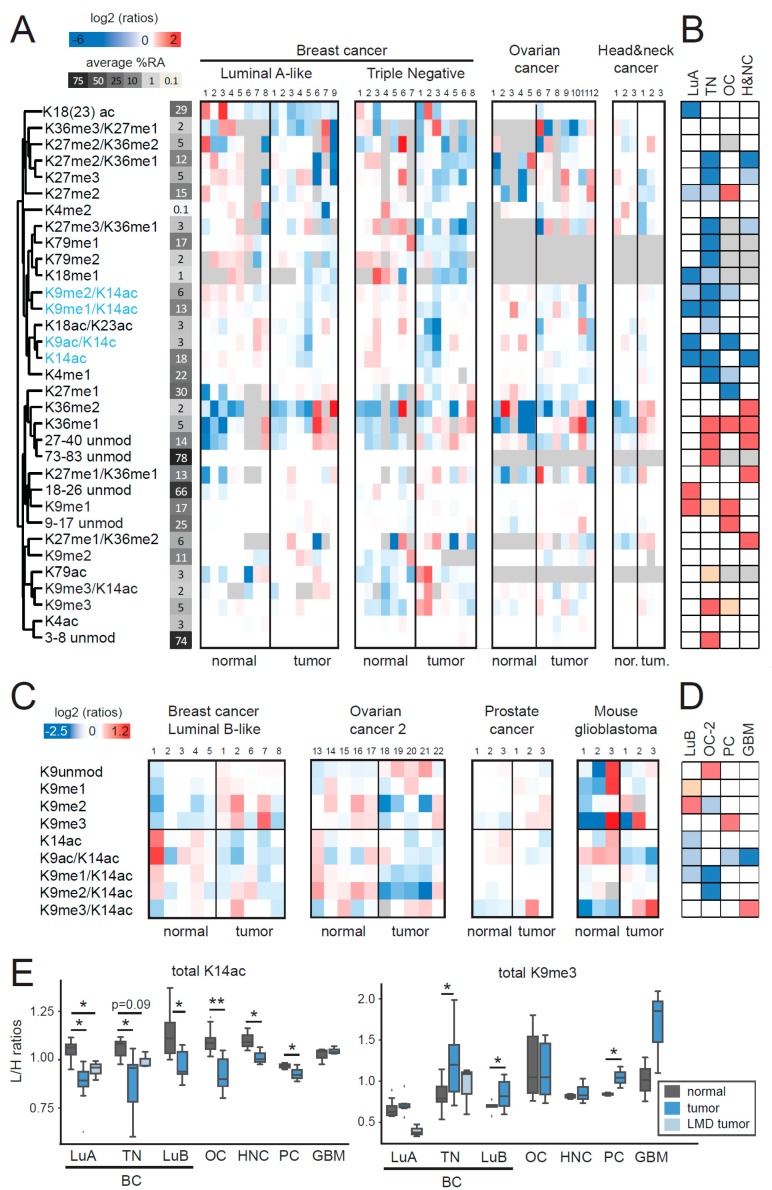
Comparison of histone H3 post-translational modifications (PTMs) in tumor and normal tissues. (**A**,**C**) Heatmap display and hierarchical clustering of the log2 transformed ratios obtained for the indicated histone H3 (H3) PTMs for different tumor types. Matched normal and tumor tissues are indicated by the same sample number. L/H (light/heavy) relative abundances ratios obtained with the super SILAC (Stable Isotope Labeling with Amino acids in Cell culture) strategy (light channel: Patient sample, heavy channel: Spike-in super-SILAC standard), normalized over the average value across the samples belonging to the same tumor type are shown. In (**C**) only the differentially modified forms of histone H3 peptides 9–17 are shown. The average % relative abundance (%RA) across samples is indicated by shades of grey and numbers on the left side (see Appendix A for a histogram representation of %RAs). The grey color indicates those peptides that were not quantified. (**B**,**D**) Modified peptides were compared by unpaired *t*-test in tumors compared with normal tissues. The red color indicates an increase in tumors, the blue color a decrease (*p* < 0.05 for darker colors, *p* < 0.1 for lighter colors). The grey color indicates those peptides that could not be quantified in formalin-fixed paraffin-embedded (FFPE) tissues or for which enough data points to obtain a p-value were not available. (**E**) Boxplot representation of the L/H ratios for total H3K14ac (given by the sum of H3K14ac, H3K9me1/K14ac, H3K9me2/K14ac, H3K9me3/K14ac, and H3K9ac/K14ac) and total H3K9me3 (given by the sum of H3K9me3 and H3K9me3/K14ac) for all the tumor types tested. For luminal A-like and triple negative breast cancers, a comparison with samples where tumor cells were isolated by laser microdissection (n = 3 for each subtype) is also shown. Normal and tumor samples were compared by *t*-test (* *p* < 0.05, ** *p* < 0.01). BC: breast cancer; LuA: Luminal A-like; LuB: Luminal B-like; TN: triple negative; OC: ovarian cancer; HNC: head and neck cancer; PC: prostate cancer; GBM: glioblastoma; LMD: laser micro-dissected.

**Figure 2 cancers-11-00723-f002:**
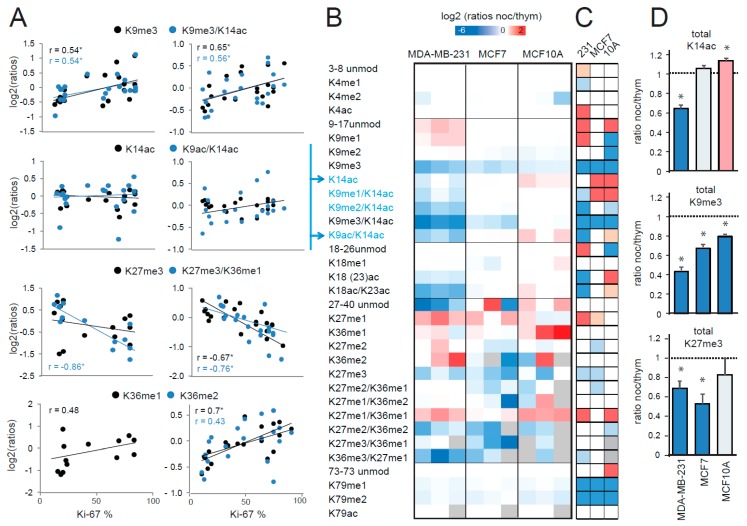
Correlation of histone post-translational modifications (PTMs) and proliferation rate. (**A**) L/H (light/heavy) ratios for the indicated peptides were plotted against the Ki-67 proliferation index for the frozen breast cancer patient samples analyzed in this study and are shown in Figure 1A, C (left panel), or breast cancer FFPE samples analyzed in [15] (right panel, containing luminal A-like, luminal B-like, HER2 (human epidermal growth factor receptor 2) positive, and triple negative subtypes). The Pearson’s correlation score (r) is shown when above 0.4 and an asterisk indicates a correlation with a *p* value < 0.05. (**B**) Heatmap display of the log2 transformed ratios obtained for the indicated histone H3 PTMs for the MDA-MB-231 and MCF7 breast cancer cell lines and the MCF10A normal breast cell line. “Ratios of ratios” are shown, which were obtained by dividing the L/H ratios for nocodazole-treated cells (synchronized in G2-M phase) by the L/H ratio for thymidine treated cells (synchronized in G1-S phase). The grey color indicates peptides that were not quantified. Peptides containing K14ac are highlighted in blue. (**C**) Modified peptides in nocodazole- and thymidine-synchronized cells were compared by paired *t*-test. The red color indicates an increase in G2-M phase, the blue color a decrease (*p* < 0.05 for darker colors, *p* < 0.1 for lighter colors). The grey color indicates peptides for which enough data points to obtain a *p*-value were not available. (**D**) Histograms representing the ratios of nocodazole- and thymidine-treated cells for total H3K14ac (given by the sum of H3K14ac, H3K9me1/K14ac, H3K9me2/K14ac, H3K9me3/K14ac, and H3K9ac/K14ac), total H3K9me3 (given by the sum of H3K9me3 and H3K9me3/K14ac), and total H3K27me3 (given by the sum of H3K27me3 and H3K27me3/K36me1) for the three cell lines tested. Changes in nocodazole-compared with thymidine-treated cells are indicated by an asterisk (*: *p* < 0.1). The red color indicates an increase in G2-M phase, while the blue color indicates a decrease.

**Figure 3 cancers-11-00723-f003:**
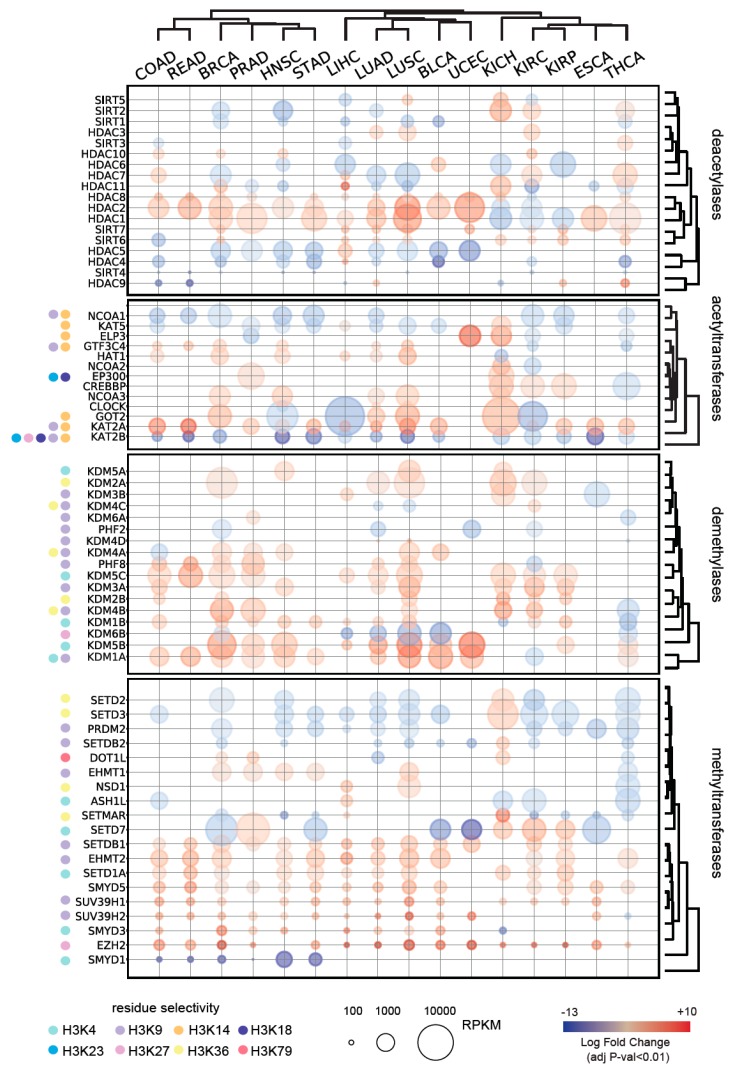
Differential expression of histone modifying enzymes in normal and tumor tissues. Only significantly deregulated genes (adjusted *p*-value < 0.01) are displayed. Histone modifying enzyme (HME) specificity for common histone H3 PTMs is marked. BLCA: Bladder Urothelial Carcinoma; BRCA: Breast invasive carcinoma; COAD: Colon adenocarcinoma; ESCA: Esophageal carcinoma; HNSC: Head and Neck squamous cell carcinoma; KICH: Kidney Chromophobe; KIRC: Kidney renal clear cell carcinoma; KIRP: Kidney renal papillary cell carcinoma; LIHC: Liver hepatocellular carcinoma; LUAD: Lung adenocarcinoma; LUSC: Lung squamous cell carcinoma; PRAD: Prostate adenocarcinoma; READ: Rectum adenocarcinoma; STAD: Stomach adenocarcinoma; THCA: Thyroid carcinoma; UCEC: Uterine Corpus Endometrial Carcinoma.

**Figure 4 cancers-11-00723-f004:**
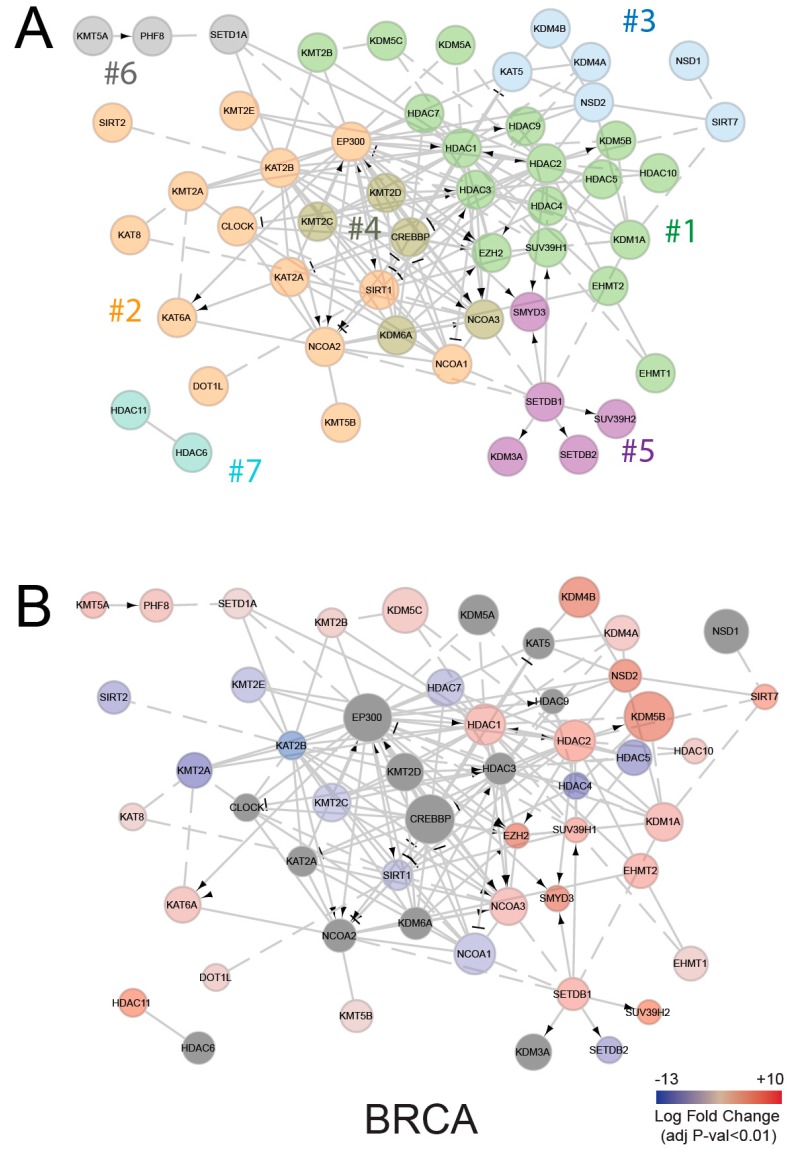
Functional interaction network of histone modifying enzymes (HMEs). (**A**) Functional interaction network of HMEs, generated through ReactomeFIViz [45] and colored on the basis of node clustering, which is achieved by optimizing network modularity. (**B**) HME interaction networks, where red and blue colors indicate up- or down-regulation in the tumors, compared with normal tissues, and node diameters are proportional to RPKM (reads per kilobase million) base mean from DESEQ2 (differential gene expression analysis based on the negative binomial distribution) analysis. BRCA; Breast invasive carcinoma.

**Figure 5 cancers-11-00723-f005:**
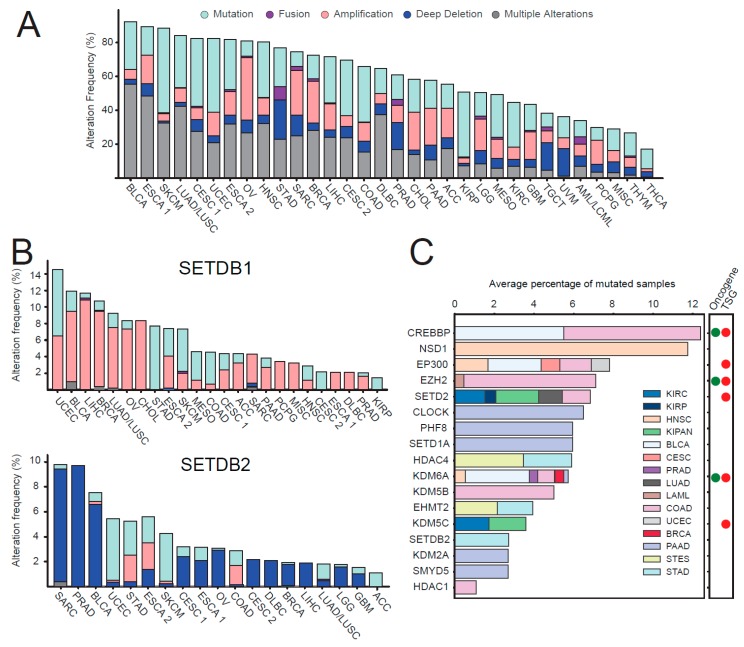
Mutational analysis of HMEs in cancer. (**A**) Stacked bar plot summarizing the frequency (minimum % of altered cases = 1) and type of mutations of HMEs in the TCGA PanCan 2018 cohort [47]. Different colors correspond to different types of mutations. (**B**) Same representation as in A for the SETDB1 and SETDB2 genes. (**C**) Horizontal stacked bar plot for significantly mutated HME genes (MutSig2CV adjusted *p*-value < 0.01). The bar widths are proportional to the number of samples mutated divided by the cohort size. When a gene is significantly mutated in multiple cancer types, the resulting cohort is given by the sum of the total samples of the individual cohorts. The right panel indicates whether the genes have been reported as oncogenes and/or tumor suppressor genes (TSG) in at least one tumor type in the Cancer Census Genes [48]. AML: Acute Myeloid Leukemia; ACC: Adrenocortical carcinoma; BLCA: Bladder Urothelial Carcinoma; LGG: Brain Lower Grade Glioma; BRCA: Breast invasive carcinoma; CESC: Cervical squamous cell carcinoma (1) and endocervical adenocarcinoma (2); CHOL: Cholangiocarcinoma; CML: Chronic Myelogenous Leukemia; COAD: Colon adenocarcinoma; ESCA: Esophageal squamous cell carcinoma (1) and esophagogastric adenocarcinoma (2); GBM: Glioblastoma multiforme; HNSC: Head and Neck squamous cell carcinoma; KIRC: Kidney renal clear cell carcinoma; KIRP: Kidney renal papillary cell carcinoma; LIHC: Liver hepatocellular carcinoma; LUAD; Lung adenocarcinoma; LUSC: Lung squamous cell carcinoma; DLBC: Lymphoid Neoplasm Diffuse Large B-cell Lymphoma; MESO: Mesothelioma; MISC: Miscellaneous; OV: Ovarian serous cystadenocarcinoma; PAAD: Pancreatic adenocarcinoma; PCPG: Pheochromocytoma and Paraganglioma; PRAD: Prostate adenocarcinoma; SARC: Sarcoma; SKCM: Skin Cutaneous Melanoma; STAD: Stomach adenocarcinoma; TGCT: Testicular Germ Cell Tumors; THYM: Thymoma; THCA: Thyroid carcinoma; UCS: Uterine Carcinosarcoma; UCEC; Uterine Corpus Endometrial Carcinoma; UVM: Uveal Melanoma.

**Figure 6 cancers-11-00723-f006:**
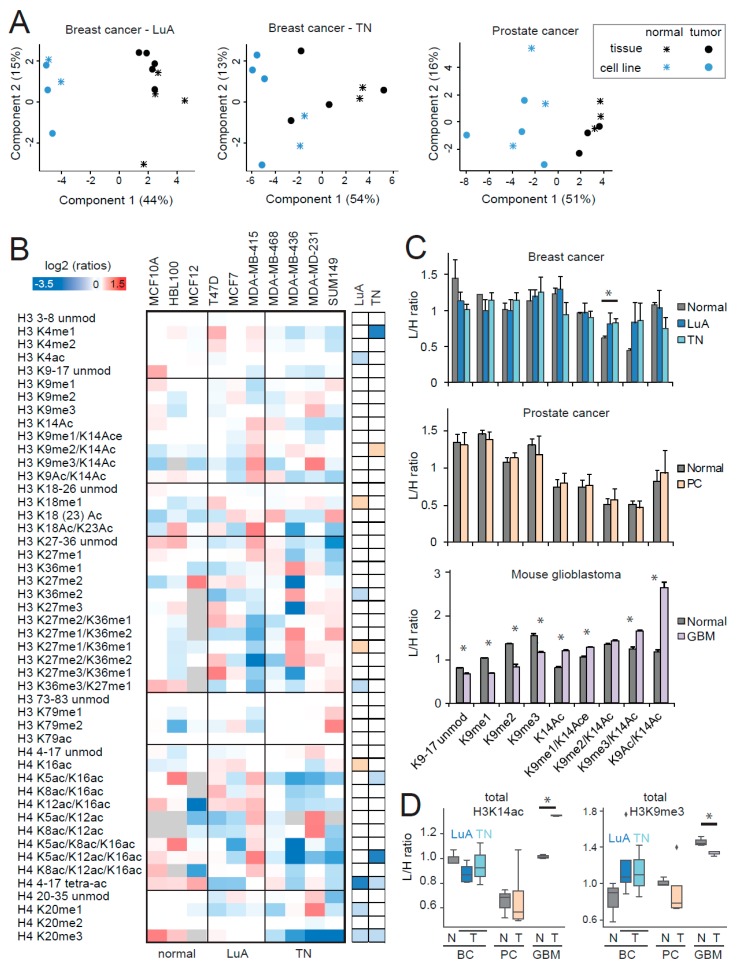
Histone post-translational modifications (PTM) profiling of normal and tumoral cell lines. (**A**) Principal component analysis of normal and tumoral tissues and cell lines for the luminal A-like and triple negative breast cancer and prostate cancer models. The grey color indicates peptides that were not quantified. (**B**) Heatmap display of the log2 transformed ratios obtained for the indicated histone PTMs in normal and tumor breast cell lines. Tumoral cell lines were divided in luminal A (LuA) and triple negative (TN) and were analyzed separately. L/H (light/heavy) relative abundances ratios obtained with the super-SILAC strategy (light channel: Normal/tumor cell line, heavy channel: Spike-in super-SILAC standard), normalized over the average value across all the samples. Some of the data for histone H3 PTMs is from [13] (see dataset S1). Right panel: Modified peptides were compared in LuA/TN tumor cells and normal cell lines by unpaired *t*-test. The red color indicates an increase in tumors, the blue color a decrease (*p* < 0.05 for darker colors, *p* < 0.1 for lighter colors). (**C**) L/H ratios for the differentially modified versions of the H3 9–17 peptide in the indicated tumor models. *: *p* < 0.05 by Student’s *t*-test. (**D**) Boxplot representation of the L/H ratios for total H3K14ac (given by the sum of H3K14ac, H3K9me1/K14ac, H3K9me2/K14ac, H3K9me3/K14ac, and H3K9ac/K14ac) and total H3K9me3 (given by the sum of H3K9me3 and H3K9me3/K14ac) for all the tumor models tested. Normal and tumor samples were compared by *t*-test (**p* < 0.05). BC: Breast cancer; PC: Prostate cancer; GBM: Glioblastoma, N: Normal; T: Tumor. In A–D, the “normal” prostate cell lines were infected with HPV.

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
