# Peer review of "Profiling of Epigenetic Features in Clinical Samples Reveals Novel Widespread Changes in Cancer"

_cancers, 2019, doi:10.3390/cancers11050723_

Round 1
Reviewer 1 Report
Noberini et al perform mass spectrometry-based analysis of histone PTMs from different types of primary samples to profile histone H3 lysine methylation and acetylation in different human cancer and normal tissues. Although some tumor specific differences are identified, there most relevant finding is the decrease of H3K14ac as a novel general hallmark of cancer. They further correlate different histone PTMs with ki67 (proliferation) and find that, while tumor changes in K27, K36 and K9 methylation may be due to increased tumor proliferation rates, H3K14 acetylation is cell cycle independent. The authors further analyze TCGA RNA-seq data to find possible correlations between their findings in PTMs analysis and levels of histone modifying enzymes and repeat their mass spectrometry-based analysis on cells lines to find that many of the histones PTMs differences observed in primary tumoral tissue are not conserved in established cell lines.
The study is relevant and the article is well written. The finding that H3K14ac is reduced in primary tumor samples is important, however, some drawbacks concerning sample number and associated extrapolations should be taken with care and discussed in the text.
- The finding that H3K14 acetylation is reduced in several primary cancer samples is very important. Although the study benefits from the analysis of primary patient samples (and not only cell lines), the samples analyzed correspond to a limited sub-set of tumor types. It is still possible that this is a characteristic of the sub-types analyzed and not a general hallmark of cancer. This should be mentioned.
- The RNA-seq analysis is performed for 16 types of cancer in TCGA to which normal/cancer matching data is available, however the initial MS analysis on PTMs was performed in a limited number of tumor types. Since differences in HMEs are most probably tumor specific or mutational background-specific it seems unlikely that a global analysis would correlate with specific changes observed in the tumor types analyzed. It could make more sense to look at HMEs expression data for the tumor types and normal tissues analyzed for MS for histone PTMs.
- The MS analysis on cell lines versus cancer samples showing that cell lines do not retain the differences in PTMs observed in tumors samples is interesting and relevant. But an analysis comparing the same tumor samples and matched derived cell lines would have to be done for a definitive conclusion. While the analysis is useful, is not ideal and that should be acknowledged.
Author Response
Noberini et al perform mass spectrometry-based analysis of histone PTMs from different types of primary samples to profile histone H3 lysine methylation and acetylation in different human cancer and normal tissues. Although some tumor specific differences are identified, there most relevant finding is the decrease of H3K14ac as a novel general hallmark of cancer. They further correlate different histone PTMs with ki67 (proliferation) and find that, while tumor changes in K27, K36 and K9 methylation may be due to increased tumor proliferation rates, H3K14 acetylation is cell cycle independent. The authors further analyze TCGA RNA-seq data to find possible correlations between their findings in PTMs analysis and levels of histone modifying enzymes and repeat their mass spectrometry-based analysis on cells lines to find that many of the histones PTMs differences observed in primary tumoral tissue are not conserved in established cell lines.
The study is relevant and the article is well written. The finding that H3K14ac is reduced in primary tumor samples is important, however, some drawbacks concerning sample number and associated extrapolations should be taken with care and discussed in the text.
The finding that H3K14 acetylation is reduced in several primary cancer samples is very important. Although the study benefits from the analysis of primary patient samples (and not only cell lines), the samples analyzed correspond to a limited sub-set of tumor types. It is still possible that this is a characteristic of the sub-types analyzed and not a general hallmark of cancer. This should be mentioned.
R: We agree with the reviewer on this observation. This aspect is now discussed in the “Conclusion” section.
The RNA-seq analysis is performed for 16 types of cancer in TCGA to which normal/cancer matching data is available, however the initial MS analysis on PTMs was performed in a limited number of tumor types. Since differences in HMEs are most probably tumor specific or mutational background-specific it seems unlikely that a global analysis would correlate with specific changes observed in the tumor types analyzed. It could make more sense to look at HMEs expression data for the tumor types and normal tissues analyzed for MS for histone PTMs.
R: We agree with the reviewer that looking at the HME expression data for the tumor and normal tissues analyzed by MS would have been ideal. However, to avoid unreliable results we performed a RNA-seq differential expression analysis only for TCGA cancer types with at least 10 tumor-normal tissue samples. It was not possible to find matched normal samples for ovarian cancer and glioblastoma models. This is now stated in the “Material and methods” section on page 18.
Question 3: The MS analysis on cell lines versus cancer samples showing that cell lines do not retain the differences in PTMs observed in tumors samples is interesting and relevant. But an analysis comparing the same tumor samples and matched derived cell lines would have to be done for a definitive conclusion. While the analysis is useful, is not ideal and that should be acknowledged.
R: We added this observation in the “Results and discussion” sections, at page 13.
Reviewer 2 Report
The manuscript presented by Noberini et al., entitled “Profiling of epigenetic features in clinical samples reveals novel widespread changes in cancer” is a well-designed study, with high technical quality. The results of the study are relevant to understand the epigenetic landscape in different tumors and the differences with cell lines. This quantification strategy may be used for the evaluation of different PTMs as markers for patient stratification or prognosis. However, there are several aspects that need to be addressed before considering its publication in Cancers:
1. The abstract must be improved as it describes the purpose of the study and methodology, but it does not highlight the key findings and conclusions of the study.
2. MS-based profiling of normal and tumor tissues.
In the methods section the heavy standard used for quantification and the digestion conditions are the same throughout the study:
- Can you comment why some peptides are missing in the heavy standard in some samples (not in FFPE samples), while readily detected in most of the samples? See dataset 1, LuA-BC and TN-BC.
- Are all the peptides detected in the samples present in the heavy standard? If not, the peptides present in tumor samples that could not be quantified must be listed.
A more detailed explanation must be included in the results of the quantification: i.e., which PTM changes are different between luminal and Triple negative cancers?
In line 130, did you mean H3 16-26 peptide or H3 18-26 peptide?
In line 138-140 is stated that in all tumor models, except in luminal A-like an increase in H3K36 dimethylation is observed, while in Figure 1B such an increase is only observed in head and neck tumors.
In figure 1A, the use of color gray in the databox (not %RA) is not explained in the legend. There are several discrepancies between figure 1E and the text (lines 169-176). In the figure the decrease in H3K14ac is significant in all models, except in glioblastoma, and the increase in total H3K9me3 is significant in three of the seven models, while in the text is stated that the decrease of H3K14ac was observed in all models and the increase of H3K9me3 is observed in four models. Which information is correct?
3. MS-based profiling of cycling cells
In the analysis of synchronized cell lines two aspects must be taken into consideration during the analysis and explained:
- It is known that core histones are mainly synthesized during S-phase. In addition, after the treatment with thymidine, 49% of MCF7 cells are in S-phase, while in the other two cell lines the percentage of cells in S-phase is ~20% or less (Figure S2). Do you consider that the amount of cells in S-phase may have an impact in the result of the PTM quantification?
- There is a difference of more than 30% in the percentage of cells in G2/M between the cell lines: MCF10A (~60%), MCF7 (~49%) and MDA-MB-231 (~90%). This difference is quite large and could influence the “ratio of ratios” shown in Figure 2B, 2C. In fact, MDA-MB-231 is the most divergent cell line of the three analyzed, and shows a decrease in four of the five peptides containing H3K14ac (Figure 2C)
4. Expression of histone modifying enzymes in normal and tumor tissues
Figure 3 shows that several acetyltransferases besides KAT2A, KAT2B and NCO1A can acetylate H3K14. Their contribution to the decrease of this PTM in tumors should be commented in the text.
In figure 3 the correspondence between the abbreviations used and the name of each cancer type should appear in the legend, and also in the materials and method section. The information of this figure would be clearer if the dendrogram is located in the upper part of the figure.
In the text, when analyzing the expression of the enzymes belonging to different clusters is stated that members of cluster 2 are generally downregulated, while members of cluster 1, 3 and 6 are upregulated. This is not entirely true, as there are many members in the clusters that are not affected in the same way. For instance, in prostate cancer (Figure S6 bottom panel) only three enzymes of cluster 2 (14 enzymes) are downregulated. For that reason, it would be more accurate to say that the majority of the downregulated enzymes belong to cluster 2 and the upregulated enzymes to clusters 1, 3 and 6.
The size of the spheres in panel 4B and figure S6 must be explained. A color scale for the intensity of the colors of the spheres, corresponding to changes in expression must be added. The name of the cancers should appear in the legend. The legend of figure S6 must also contain the details present in the legend of 4B.
5. Comparison of histone PTMs in normal and tumor cell lines
The results of the profiling of histone PTMs in cell lines, shows great variability in the quantification, especially in peptides with more than one modification. To better sustain the conclusions of the manuscript, two new panels (similar to Figure 1E) must be added in Figure 5, representing the variation of specific modifications considering all the peptides detected containing that PTM, by cancer type, and also in each cell line. Perhaps the second analysis may be useful to detect which cell line better retain the epigenetic patterns of the primary tumor.
In this section, the results obtained with mouse glioblastoma (GBL) primary neurosphere cultures are poorly commented. It is by far the model where more peptides have significant differences in contrast to normal cells. In this model there is a significant increase in H3K9me3, confirming the results in tumors. It was also detected a significant increase in two peptides with H3K14ac (contrary to the decrease observed in tumors), while the third peptide with this modification is decreased in GBL, albeit with no statistical significance. It should be taken into consideration that in glioblastoma tumor samples there were no significant differences in H3K4ac (Fig. 1E).
Author Response
The manuscript presented by Noberini et al., entitled “Profiling of epigenetic features in clinical samples reveals novel widespread changes in cancer” is a well-designed study, with high technical quality. The results of the study are relevant to understand the epigenetic landscape in different tumors and the differences with cell lines. This quantification strategy may be used for the evaluation of different PTMs as markers for patient stratification or prognosis. However, there are several aspects that need to be addressed before considering its publication in Cancers:
R: We thank the reviewer for the positive evaluation of the significance of our findings. We tried to address his/her comments, as detailed below.
1. The abstract must be improved as it describes the purpose of the study and methodology, but it does not highlight the key findings and conclusions of the study.
R: We modified the abstract to better convey the results of our study.
2. MS-based profiling of normal and tumor tissues.
In the methods section the heavy standard used for quantification and the digestion conditions are the same throughout the study:
- Can you comment why some peptides are missing in the heavy standard in some samples (not in FFPE samples), while readily detected in most of the samples? See dataset 1, LuA-BC and TN-BC.
- Are all the peptides detected in the samples present in the heavy standard? If not, the peptides present in tumor samples that could not be quantified must be listed.
R: All the peptides detected in the samples should be present in the heavy standard. However, occasionally, due to variability of the MS runs, low abundance peptides may not be detected. The few peptides that could not be quantified in specific samples due to their absence in the internal standards have been highlighted in Datasets S1-S2.
A more detailed explanation must be included in the results of the quantification: i.e., which PTM changes are different between luminal and Triple negative cancers?
R: We added a more detailed description of the results from Figure 1 on pages 3-4.
In line 130, did you mean H3 16-26 peptide or H3 18-26 peptide?
R: We meant H3 18-26. We thank the reviewer for pointing out this mistake.
In line 138-140 is stated that in all tumor models, except in luminal A-like an increase in H3K36 dimethylation is observed, while in Figure 1B such an increase is only observed in head and neck tumors.
R: With the sentence “increase of mono- and di-methylation on H3K36” we intended an increase of either one of the forms. Because we agree that this was not clear, we changed the sentence to “the methylated forms of H3K36”, to include both H3K36me1 and H3K36me2.
In figure 1A, the use of color gray in the databox (not %RA) is not explained in the legend.
R: We added this piece of information in Figure 1A, as well as Figures 2B and 5B.
There are several discrepancies between figure 1E and the text (lines 169-176). In the figure the decrease in H3K14ac is significant in all models, except in glioblastoma, and the increase in total H3K9me3 is significant in three of the seven models, while in the text is stated that the decrease of H3K14ac was observed in all models and the increase of H3K9me3 is observed in four models. Which information is correct?
R: The sentence “The decrease of H3K14ac-containing peptides was confirmed in all these tumor samples” (lines 173-174 of the revised manuscript) referred to Figure 1C-D, where indeed all tumor models showed a decrease in at least one form containing H3K14ac. Because we agree that this sentence might have been misleading, we have added a specific reference to the correct panel, and mentioned on lines 180-181 that we did not observe a decrease of total H3K14ac, but only of H3K14ac in combination with H3K9ac, in mouse glioblastoma. We corrected in the text the number of tumor models where we could observe a significant increase of H3K9me3 (which is three). We thank the reviewer for pointing out the mistake.
3. MS-based profiling of cycling cells
In the analysis of synchronized cell lines two aspects must be taken into consideration during the analysis and explained:
- It is known that core histones are mainly synthesized during S-phase. In addition, after the treatment with thymidine, 49% of MCF7 cells are in S-phase, while in the other two cell lines the percentage of cells in S-phase is ~20% or less (Figure S2). Do you consider that the amount of cells in S-phase may have an impact in the result of the PTM quantification?
R: We agree with the reviewer’s remark that the different proportion of cells in S-phase in different cell lines treated with thymidine may potentially have an impact in the result of the PTM quantification. However, although the proportion of cells in S-phase was different in MCF7 compared with the other two cell lines, the histone PTMs profile of MCF7 and MCF10A cells were very similar. Therefore, we assumed that the different percentage of cells in S-phase does not substantially impact on PTM quantification.
- There is a difference of more than 30% in the percentage of cells in G2/M between the cell lines: MCF10A (~60%), MCF7 (~49%) and MDA-MB-231 (~90%). This difference is quite large and could influence the “ratio of ratios” shown in Figure 2B, 2C. In fact, MDA-MB-231 is the most divergent cell line of the three analyzed, and shows a decrease in four of the five peptides containing H3K14ac (Figure 2C)
R: It is true that the percentage of cells in G2/M phase varies depending on the cell lines, which could affect the “ratio or ratios”. Nevertheless, we believe that this should not alter the overall result of the analysis, given that the percentage of cells in G2/M is substantially larger in cells treated with nocodazole compared to the ones treated with thymidine for all the cell lines tested. A lower percentage of cells in G2/M could decrease the amplitude of the “ratio of ratio”, but not give opposite results, as is it the case for the H3K14ac mark (which decreased in G2-M phase in MDA-MB-231, but remained constant in MCF7 and even increased in MCF10A). This aspect is now mentioned in the “Material and methods” section, at page 16.
To better visualize the result of this analysis, we showed in Figure 2D the nocodazole/thymidine ratio for total H3K14ac (given by the sum of H3K14ac, H3K9Me1/14ac, H3K9Me2/14ac, H3K9ac/14ac), total H3K9me3 (given by the sum of H3K9Me3, H3K9Me3/14ac), and H3K27me3 (given by the sum of H3K27me3 and H3K27m3/K36me2).
4. Expression of histone modifying enzymes in normal and tumor tissues
Figure 3 shows that several acetyltransferases besides KAT2A, KAT2B and NCO1A can acetylate H3K14. Their contribution to the decrease of this PTM in tumors should be commented in the text.
R: We added a sentence mentioning these additional enzymes on page 9. We thank the reviewer for the suggestion.
In figure 3 the correspondence between the abbreviations used and the name of each cancer type should appear in the legend, and also in the materials and method section. The information of this figure would be clearer if the dendrogram is located in the upper part of the figure.
R: We added the names of the cancer types both in the legend for Figure 3 and in the Material an Method section, and moved the dendrogram to the upper part of the figure.
In the text, when analyzing the expression of the enzymes belonging to different clusters is stated that members of cluster 2 are generally downregulated, while members of cluster 1, 3 and 6 are upregulated. This is not entirely true, as there are many members in the clusters that are not affected in the same way. For instance, in prostate cancer (Figure S6 bottom panel) only three enzymes of cluster 2 (14 enzymes) are downregulated. For that reason, it would be more accurate to say that the majority of the downregulated enzymes belong to cluster 2 and the upregulated enzymes to clusters 1, 3 and 6.
R: We modified the text according to the suggestion of the reviewer.
The size of the spheres in panel 4B and figure S6 must be explained. A color scale for the intensity of the colors of the spheres, corresponding to changes in expression must be added. The name of the cancers should appear in the legend. The legend of figure S6 must also contain the details present in the legend of 4B.
R: We modified Figure 4B and S6, and the corresponding legends according to the revewer’s useful suggestions.
5. Comparison of histone PTMs in normal and tumor cell lines
The results of the profiling of histone PTMs in cell lines, shows great variability in the quantification, especially in peptides with more than one modification. To better sustain the conclusions of the manuscript, two new panels (similar to Figure 1E) must be added in Figure 5, representing the variation of specific modifications considering all the peptides detected containing that PTM, by cancer type, and also in each cell line. Perhaps the second analysis may be useful to detect which cell line better retain the epigenetic patterns of the primary tumor.
R: As suggested by the reviewer, we have added to figure 6 (olf figure 5) a panel (D) showing the average variation of H3K14ac and H3K9me3 considering all the peptides detected containing those modifications. For figure clarity and simplicity, and because our main goal in this part of the study was to verify whether normal-tumor changes were retained in culture, we did not show individual cell lines. The issue of histone PTMs patterns in culture compared with primary tissues has been addressed in a comprehensive manner in our previous publication (Noberini et al., Extensive and systematic rewiring of histone post-translational modifications in cancer model systems. Nucleic Acids Res. 2018), where we also indicated which are the breast cancer and glioblastoma cell lines most representative of the tissue of origin, based on histone PTM patterns.
In this section, the results obtained with mouse glioblastoma (GBL) primary neurosphere cultures are poorly commented. It is by far the model where more peptides have significant differences in contrast to normal cells. In this model there is a significant increase in H3K9me3, confirming the results in tumors. It was also detected a significant increase in two peptides with H3K14ac (contrary to the decrease observed in tumors), while the third peptide with this modification is decreased in GBL, albeit with no statistical significance. It should be taken into consideration that in glioblastoma tumor samples there were no significant differences in H3K4ac (Fig. 1E).
R: We agree with the reviewer that the results from the GBL mouse model were poorly commented, so we added several considerations regarding this aspect, on page 13. Please note that during the revision we realized that there were minor mistakes in the glioblastoma cell lines panel (Figure 5C, bottom) and the heatmap in Figure 5B, which we have therefore replaced. We would like to point out that this correction does not not alter the overall results of the analysis.
Reviewer 3 Report
In this manuscript, Bonaldi and coworkers profiled histone H3 PTM by mass spectrometry in a panel of cancer models. PTM differences between normal and cancer tissues were compared and revealed some features of PTM in a variety of cancer types. I suggest his manuscript can be accepted for publication in Cancers after revision.
1. Page 2, line 76 "By demonstrating for the first time that..." sounds exaggerated since a lot of investigations are into the correlation between PTM and cancers.
2. Are the tumoral and normal tissue from the same patient? How to quantify the area of tissues adjacent to the tumor are normal?
3. Primary tumors are highly heterogeneous, how to determine which portion of the tumor should be used in the PTM profiling?
4. There are two sections numbered "2.3".
5. There are few typos for "μm", please check throughout the manuscript.
6. What is the full name of "SILAC"? All acronyms in the manuscript should be checked.
Author Response
In this manuscript, Bonaldi and coworkers profiled histone H3 PTM by mass spectrometry in a panel of cancer models. PTM differences between normal and cancer tissues were compared and revealed some features of PTM in a variety of cancer types. I suggest his manuscript can be accepted for publication in Cancers after revision.
1. Page 2, line 76 "By demonstrating for the first time that..." sounds exaggerated since a lot of investigations are into the correlation between PTM and cancers.
R: We removed “for the first time” from the sentence.
2. Are the tumoral and normal tissue from the same patient? How to quantify the area of tissues adjacent to the tumor are normal?
R: As mentioned on page 3 (line 112), page 15 (line 480) and in Tables S1-S4, normal and tumor samples were matched, when possible. Matched normal and tumor tissues are indicated by the same sample number. As now explicitly mentioned on page 15 (line 489), all normal and tumor samples were selected and evaluated by a trained pathologist.
3. Primary tumors are highly heterogeneous, how to determine which portion of the tumor should be used in the PTM profiling?
R: As mentioned on page 13, the criteria for the selection of the samples to be analyzed were a tumor cellularity of at least 50%, and the absence of in situ carcinoma areas, large necrosis areas and massive flogistic infiltrate. Therefore, except for the analysis of laser microdissected tumor populations shown in Figure 1E, in this study we did not take into account tumor heterogeneity. However, tissue heterogeneity is an important issue that will have to be addressed in the future, and that we now discuss in the “Conclusions” section at page 19.
4. There are two sections numbered "2.3".
R: We corrected the mistake. We thank the reviewer for pointing it out.
5. There are few typos for "μm", please check throughout the manuscript.
R: We corrected the typos.
6. What is the full name of "SILAC"? All acronyms in the manuscript should be checked.
R: We added the full name for SILAC, and checked all the acronyms.
Reviewer 4 Report
This study is descriptive and would benefit from improved focus aside from the associations that emerged. The introduction would benefit from greater detail, more discussion in relation to the existing literature and setting the scene for a greater focus.
It is not clear in some places in the results how the methods were implemented. Conclusions are rather general and not discussed in relation to the existing literature. Supporting functional studies would have been a benefit.
Use of English in places is rather general, with some minor errors and significant use of sweeping statements that are not strongly supported or discussed in relation to the existing literature. In places it would be better to be more quantitative / analytical/ critical.
For example:
Line 49 “aberrations in the epigenetic information can cause dramatic changes in the phenotype of a cell” – what constitutes a dramatic change and what is its nature?
Line 51 “Histone modifications represent a vast catalogue of combinatorial events” -vast is imprecise, there are a known number of modifications and a predictable number of combinations.
Tumour cell cellularity of > 50% were reported for the comparison of Histone PTMs in the tumour tissues and the latter were obtained in an ‘ad hoc’ leaving open the possibility that other cell types for example infiltrating immune cells, an inflammatory response or methods of tissue collection could be having an effect on the results observed. Even for tumours with higher levels of proliferating cells, the latter could still represent a small proportion of the total and may not be significant. Nevertheless, proliferation is focused on as a possible explanation, without considering other possibilities like relative levels of differentiation. These could have been taken into account or discussed, for example the basal versus ER/PR breast cancers.
When cells in G1/S versus G2/M were compared the text suggested that these were arrested cells, but the methods left it open to the possibility that the cells were synchronized and in cycle. This needs to be clarified. If the cells were arrested, how effective was this because arrest could also be expected to have an effect on PTMs. At what time point were samples taken?
Ideally, functional studies would have been performed for example to show the consequences of changing the levels of enzymes responsible for PTMs on the modifications observed. Their expression differences may not reflect net activity; their mutation, turnover or relative proportions could all be consequential. How many of the expression data points in the TCGA data referred to enzymes that were mutated? Greater use of hypothesis testing could have been made.
Significant associations were observed but multiple testing was extensive and at least some of the significance could have been by chance. Correction for multiple testing should have been considered or its absence justified. The most consistent significance was observed for the mouse glioblastoma model, and it would have better if more had been made of this.
In summary, a more penetrating analysis and resultant discussion was possible even using only the existing data and ideally greater use of hypothesis testing and follow up (functional) studies would have been performed.
Author Response
This study is descriptive and would benefit from improved focus aside from the associations that emerged. The introduction would benefit from greater detail, more discussion in relation to the existing literature and setting the scene for a greater focus.
It is not clear in some places in the results how the methods were implemented. Conclusions are rather general and not discussed in relation to the existing literature. Supporting functional studies would have been a benefit.
Use of English in places is rather general, with some minor errors and significant use of sweeping statements that are not strongly supported or discussed in relation to the existing literature. In places it would be better to be more quantitative / analytical/ critical.
For example:
Line 49 “aberrations in the epigenetic information can cause dramatic changes in the phenotype of a cell” – what constitutes a dramatic change and what is its nature?
R: We changed the sentence to “The disruption of epigenetic mechanisms can lead to altered gene expression and cellular transformation, which play a crucial role during the initiation and progression of cancer”.
Line 51 “Histone modifications represent a vast catalogue of combinatorial events” -vast is imprecise, there are a known number of modifications and a predictable number of combinations.
R: The number of known histone modifications is constantly increasing, thanks to the continuous discovery of novel sites, combinations and even novel types of modifications, for instance histone serotonylation (doi: 10.1038/s41586-019-1024-7) just to name one. Therefore, the use of a generic term such as “vast” was intentional and we would not like to change/modify. However, to address the reviewer’s coment and better explain the fact that the number of histone PTM is expanding, we added a sentence on page 2, line 53-55.
Tumour cell cellularity of > 50% were reported for the comparison of Histone PTMs in the tumour tissues and the latter were obtained in an ‘ad hoc’ leaving open the possibility that other cell types for example infiltrating immune cells, an inflammatory response or methods of tissue collection could be having an effect on the results observed. Even for tumours with higher levels of proliferating cells, the latter could still represent a small proportion of the total and may not be significant. Nevertheless, proliferation is focused on as a possible explanation, without considering other possibilities like relative levels of differentiation. These could have been taken into account or discussed, for example the basal versus ER/PR breast cancers.
R: We completely agree with the reviewer that changes in the cell cycle may be one of many factors potentially influencing the changes in histone PTM patterns that we observed in our study, and that the presence of non-tumoral cells may have an effect on the results observed. We added this aspect in the “Conclusion” section at page 19, where we discussed potential limitations of our study and aspects to be addressed in the future to confirm and validate our finding.
When cells in G1/S versus G2/M were compared the text suggested that these were arrested cells, but the methods left it open to the possibility that the cells were synchronized and in cycle. This needs to be clarified. If the cells were arrested, how effective was this because arrest could also be expected to have an effect on PTMs. At what time point were samples taken?
R: The cells were synchronized in different phases of the cell cycle and not arrested, as correctly described in the method section. We changed the word “arrested” to “synchronized” throughout the manuscript. We thank the reviewer for pointing out this mistake.
Ideally, functional studies would have been performed for example to show the consequences of changing the levels of enzymes responsible for PTMs on the modifications observed. Their expression differences may not reflect net activity; their mutation, turnover or relative proportions could all be consequential. How many of the expression data points in the TCGA data referred to enzymes that were mutated? Greater use of hypothesis testing could have been made.
R: We agree with the observation of the reviewer that than enzyme level may not reflect their activity. Therefore, we expanded the discussion of factors that can contribute to histone PTM levels on page 9, and added a whole section of the manuscript describing the analysis of mutations found in HMEs in different tumor types. These results are described on page 11, and shown in two new figures (Figure 5 and Figure S7). We also checked for any possible relationship between mutation status and differential expression of HMEs. We derived the list of significantly mutated genes through the MutSig2CV approach (Lawrence et al., Nature, 2013; doi: 10.1038/nature12213), available from firebrowse (http://firebrowse.org/). We then checked whether significantly regulated HMEs (DESEQ2 adj Pvalue < 0.01) were also significantly mutated (MutSig2CV adj Pvalue < 0.01) through a Fisher’s exact test, finding no significant relationship (see Table SXb). We thank the reviewer for the suggestion.
Significant associations were observed but multiple testing was extensive and at least some of the significance could have been by chance. Correction for multiple testing should have been considered or its absence justified.
R: Although, as the reviewer pointed out, some significant changes could heve been seen by chance, we chose not to apply any correction for multiple testing, given the limited number of samples analyzed and the exploratory nature of this study. We now specify the absence of the correction for multiple testing in the “Statistical analysis” paragraph. In addition, we added the p-values for every comparison made in Datasets S1 and S2, following what suggested in the paper by Althouse,“Adjust for Multiple Comparisons? It’s Not That Simple”, Ann Thorac Surg., 2016.
The most consistent significance was observed for the mouse glioblastoma model, and it would have better if more had been made of this.
R: We agree with the reviewer that the results from the GBL mouse model were poorly commented, and we added several considerations regarding this aspect, on page 13. Please note that during the revision we realized that there were minor mistakes in the glioblastoma cell lines panel (Figure 5C, bottom) and the heatmap in Figure 5B, which we replaced. This change, however, does not alter the overall results of this analysis.
In summary, a more penetrating analysis and resultant discussion was possible even using only the existing data and ideally greater use of hypothesis testing and follow up (functional) studies would have been performed.
R: Following the helpful comments of the reviewers, we expanded both the data anlysis and the discussion of the results and their limits. We believe that the manuscript is now improved, and we hope that the reviewer will now judge it acceptable for publication.
Round 2
Reviewer 2 Report
The revised version of the manuscript entitled “Profiling of epigenetic features in clinical samples reveals novel widespread changes in cancer” by Noberini et al. is significantly improved. The authors have addressed all the comments to the initial version of the manuscript successfully, and therefore, I recommend its publication in Cancers.